# First report of canine Chagas disease on the Caribbean Island of Trinidad

Rod Suepaul[1], Azad Mohammed[2], Nicole L Gottdenker[3], Indira Pargass[1], Christopher Oura[1], Adesh Ramsubhag[2], Lana Gyan[4], Vrijesh Tripathi[5], Jennifer K Peterson[6]/+

[1]University of the West Indies, School of Veterinary Medicine, St. Augustine, Trinidad and Tobago
[2]University of the West Indies, Department of Life Sciences, St. Augustine, Trinidad and Tobago
[3]University of Georgia, College of Veterinary Medicine, Department of Veterinary Pathology, Athens, GA, USA
[4]Ministry of Agriculture Lands and Fisheries, Animal Production and Health Division, Champs Fleurs, Trinidad and Tobago
[5]University of the West Indies, Department of Mathematics and Statistics, Trinidad and Tobago
[6]University of Delaware, College of Agriculture and Natural Resources, Department of Entomology & Wildlife Ecology, Newark, DE, USA

**BACKGROUND** Chagas disease (CD) is a vector-borne infection caused by *Trypanosoma cruzi*, a kinetoplastid parasite of mammals. *T. cruzi* is transmitted by triatomine bugs throughout the Americas and some Caribbean islands. On the Caribbean island of Trinidad, *T. cruzi* has been isolated from triatomine bugs in several residential areas where dogs are a common pet. However, canine *T. cruzi* infection in Trinidad has never been studied.

**OBJECTIVES** We aimed to demonstrate that canine CD does occur in Trinidad through a review of veterinary records from the years 2008-2023.

**METHODS** We reviewed 3,923 case reports from Trinidad veterinary clinics for canine Chagas cases diagnosed through histological evaluation, necropsy, blood smear evaluation, and/or polymerase chain reactions (PCR).

**FINDINGS** We identified 13 confirmed and two suspected canine CD cases. Animal ages ranged from five weeks to 14 years old, with four (27%) being less than one year old, including the pup of a *T. cruzi*-infected dam. Breed varied, although one-third (5/15) were hounds. Clinical signs ranged from asymptomatic (43%; 6/14) to severely ill with limb paresis (21%; 3/14). Seven of the fifteen (47%) dogs died, and three more (20%) were euthanized. Myocarditis with visible amastigote forms were found in two-thirds (9/15) of dogs.

**MAIN CONCLUSIONS** Our findings highlight a need for increased awareness of CD among dog owners and veterinarians in Trinidad.

Key words: canine Chagas disease - *Trypanosoma cruzi* - triatomine bugs - Trinidad and Tobago

*Trypanosoma cruzi* is a vector-borne parasite of mammals, and the causative agent of Chagas disease (CD) in humans and dogs.[1,2,3] The arthropod vectors of *T. cruzi* are hematophagous insects called triatomine bugs that transmit the parasite stercorally (*i.e.*, in their excrement and not in their saliva). Insectivorous animals, including dogs, can also acquire *T. cruzi* orally through the consumption of *T. cruzi*-infected triatomine bugs, which is more efficient than stercoral transmission and can cause severe, acute disease.[4,5,6,7] Other modes of *T. cruzi* transmission in dogs include transplacental transmission from mother to fetus.

Vector-borne *T. cruzi* transmission is broadly classified into domestic and sylvatic cycles. Domestic *T. cruzi* transmission occurs near or inside human dwellings, with the parasite circulating through triatomine bug vectors, domestic and synanthropic mammals, and humans.[8] Sylvatic *T. cruzi* transmission occurs among mammals living in wild ecotopes and does not involve humans.[9] There are also peri-domestic cycles, which involve some degree of overlap between domestic and sylvatic cycles,

and involve wild mammalian species that thrive in human-dominated environments, such as raccoons and opossums.[10,11,12] Domestic dogs are considered sentinels for human CD in domestic environments, due to their close relationship with humans and their propensity to eat insects, including triatomine bugs.[13,14,15,16,17] Dogs that sleep outdoors and accompany their owners when hunting wild animals are especially vulnerable to *T. cruzi* infection.[17] Kennels or shelters housing multiple dogs in regions with triatomine bugs also pose an increased risk of canine *T. cruzi* infection.[14,18,19,20,21]

The progression of canine CD follows a similar pattern to the disease in humans, comprising acute, indeterminate, and chronic phases.[1,3,22,23] The acute phase occurs shortly following infection. Clinical signs of the acute phase include diarrhea, lethargy, swollen lymph nodes, fluid retention, ascites, and in severe cases, acute myocarditis and sudden death.[4,24,25] Dogs that recover from acute infection move into an asymptomatic indeterminate phase with a subpatent parasitemia.[26] The duration of the indeterminate phase is variable; some dogs will remain in this phase indefinitely, while others will

+ Corresponding author: jkp@udel.edu | (ID) https://orcid.org/0000-0002-0274-6143

**Handling editor:** Adeilton Alves Brandão | (ID) https://orcid.org/0000-0001-5877-607X

progress to the chronic phase within an estimated eight to 36 months.[22] Clinical signs of the chronic phase include cardiac arrhythmias and signs of cardiac compromise or heart failure, including lethargy, fainting, fluid buildup in the abdomen or lungs. Clinical signs in dogs can vary with age, with young puppies exhibiting more severe symptoms. Sudden death can occur in both the acute and chronic phases.[1]

Pathologic changes in dogs with CD include myocarditis, hepatomegaly, ascites, and cardiac enlargement and dilatation.[1,22,24,27] Megaoesophagus and megacolon may also be present in both acute and chronic phases.[28] Histologic changes include intracytoplasmic amastigote pseudocysts accompanied by lymphoplasmacytic to histiocytic inflammation, cardiomyocyte necrosis, and replacement fibrosis.[1,25,26,27,29,30] T. cruzi-infected hearts without detectible amastigotes may still yield positive results in polymerase chain reactions (PCRs) targeting T. cruzi DNA, as the parasite is not always readily detectable upon histologic examination of chronically infected dogs.[23,26,31]

There is currently no canine vaccine against CD, although development efforts are underway.[23,32] Antiparasitic medication for canine CD has been explored with mixed results, and treatment of the disease generally consists of symptom management; there is no cure for chronic canine CD.[23,31,33] Prognosis and survival of T. cruzi-infected dogs varies and not all infected dogs develop disease.[23,25,34,35]

*Chagas disease in Trinidad* - Although well-studied in South and Central America, CD is overlooked in the Caribbean Islands, despite evidence of triatomine bugs and *T. cruzi* in several of the islands.[36,37,38,39,40,41] On the island of Trinidad in the dual island nation of Trinidad and Tobago, there are six described triatomine bug species. The most common triatomine species is *Panstrongylus geniculatus* (Fig. 1; reviewed in Suepaul 2025).[41] A recent survey of *P. geniculatus* found in and around

households in Trinidad found *T. cruzi* infection in over 80% of specimens tested.[38] Recent detection of two triatomine bug species in Tobago indicates that *T. cruzi* likely circulates there as well, although studies are needed to confirm its presence.[41]

In Trinidad, as in many regions, dogs are commonly an integral member of human households, serving purposes that include companionship, hunting, and protection. Dogs from households in Trinidad that are located near triatomine bug nidi and dogs that are used for hunting wildlife reservoirs of *T. cruzi,* including nine-banded armadillos (*Dasypus novemcinctus*) and opossums (*Didelphis marsupialis*) could be at an increased risk of acquiring *T. cruzi* infection.[42,43] However, local canine *T. cruzi* infection remains unexamined. Therefore, we investigated canine CD in the island of Trinidad using 3,923 historical case records from two veterinary diagnostic clinics. Our objectives were to (i) demonstrate that *T. cruzi* transmission occurs in dogs in Trinidad; (ii) document the disease and its pathology in locally transmitted cases; (iii) raise awareness of canine CD in TT to facilitate the detection of suspected cases; (iv) provide baseline data to guide future research of canine *T. cruzi* infections in Trinidad and Tobago.

## MATERIALS AND METHODS

*Historical records search* - We conducted a retrospective study of canine necropsy and blood submission records from the University of the West Indies School of Veterinary Medicine Pathology Laboratory, as well as necropsy records from the Trinidad and Tobago Ministry of Agriculture Veterinary Diagnostic Laboratory (VDL). We searched all available digital diagnostic records (N = 3,023) of dogs submitted to the VDL between January 2008 and December 2023. Canine necropsy records were examined for myocarditis and other lesions or diagnostic results consistent with *T. cruzi* infection. Clinical records were also screened for cases where the submitting veterinarian had tested blood for *T. cruzi*, which is not routinely performed otherwise. All cases were likely locally acquired, as no history of travel outside the country was reported.

*Necropsy procedures* - Necropsy procedures reported in each record consisted of routine gross examination of all internal organs except for the central nervous system, which was only examined when indicated by the clinical history. Samples of any lesions detected, as well as representative sections of liver, lung, and kidneys were placed in 10% buffered formalin. After a 48-hour fixation, the tissues were trimmed and embedded in paraffin blocks. Sections 4 μm thick were placed in glass slides, deparaffinized, and routinely stained with hematoxylin and eosin for microscopic examination.

*Polymerase chain reaction detection methods* - In cases where PCR was used to detect *T. cruzi* infection, the assay consisted of DNA extracted from each sample (blood or representative organ) amplified in a PCR using the TcZF/R primers, which amplify a 182 bp region of *T. cruzi* nuclear DNA (5'-GCTCTTGCCCACAAGGGTGC-3' and 5'-CCAAGCAGCGGATAGTTCAGG-3').[44,45]

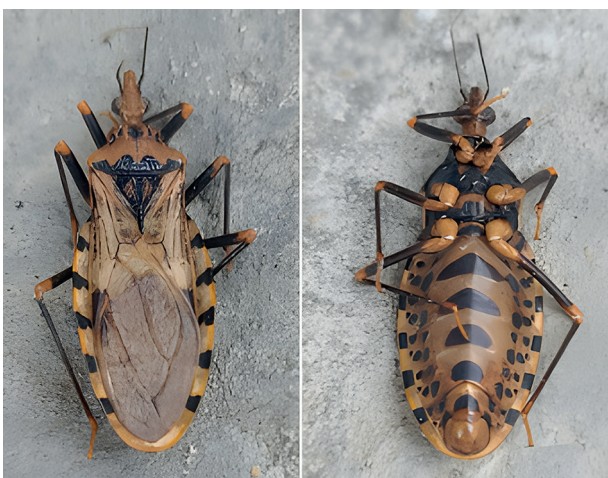

Fig. 1: *Panstrongylus geniculatus*, the main *Trypanosoma cruzi* vector in Trinidad and Tobago. Photo taken by Dr Amy Deacon and originally published in Suepaul et al.[42]

## RESULTS

*Overview* - Out of the 3,923 canine records reviewed, we found two suspected and 13 confirmed cases of canine CD. Dog age ranged from five weeks to 14 years old, with four (27%) being less than one year old, including the five-week-old pup of a *T. cruzi*-infected dam. Dog breed varied, although one-third (5/15) were hounds. Clinical signs ranged from asymptomatic (43%; 6/14) to severely ill with limb paresis (14%; 2/14). Seven of the fifteen (47%) dogs died and three more (20%) were euthanized. Myocarditis with visible amastigote forms were found in two-thirds of dogs (9/15).

*Case descriptions* - The following is a summary of the cases identified, presented in in chronological order. A complete and detailed list of case descriptions and corresponding pathology is available in Table.

*2008-2018* - The first canine CD cases identified in this review occurred in 2008 and 2013 and involved a four-year-old male Dalmatian and a nine-month-old female Rottweiler, respectively. Both died suddenly. Histological examination revealed that both animals had a severe lymphoplasmacytic and histiocytic myocarditis with visible protozoal amastigotes within the cytoplasm of cardiac myocytes, leading the diagnosing veterinarians to suspect *T. cruzi* infection. These are the only cases where PCR-based diagnostics were not done to confirm the diagnosis.

The third and fourth cases occurred in 2018 when a five-week-old male Cocker Spaniel died two days after beginning treatment for both *Rickettsia* bacterial infection and hookworm. Necropsy revealed myocarditis and intracytoplasmic protozoal amastigotes in cardiac myocytes (Fig. 2), and blood from both the pup and its mother (who was not visibly sick) tested positive for *T. cruzi* via PCR.

*2019-2023* - Diagnosed canine Chagas cases increased during this period, likely due to increased awareness of *T. cruzi* infection among veterinary clinic staff, as follows:

- The fifth case was from 2019, where an 18-month-old male foxhound was brought in with anorexia, marked weakness, and hindlimb paresis [Fig. 3 and **Supplementary data** (video)]. The owner decided to euthanize the dog. *T. cruzi* forms were observed in blood smears from the dog, indicative of acute and recent infection. Histological examination revealed myocarditis and intracytoplasmic amastigotes; DNA from heart and spleen tissues were PCR-positive for *T. cruzi*. Given the presence of *T. cruzi* amastigotes in the heart and PCR-positivity of the heart and spleen, it is highly likely that the myelitis detected was a result of *T. cruzi* infection and the cause of the hindlimb paresis in the animal. However, direct evidence of *T. cruzi* in the spinal cord was not collected, as PCR was done just on the heart and the spleen tissues and immunohistochemistry analyses were not carried out.

- The following year (2020), three adults and two pups under six months old were diagnosed with canine CD (cases 6-10, Table). Two of the dogs were hunting dogs, one of which was from a pack specializing in armadillo hunting.

Armadillos are a *T. cruzi* reservoir species and a common host for *P. geniculatus*, the most common triatomine bug species in Trinidad.[46,47] The dog, which died suddenly after displaying respiratory distress, had pathological signs of histiocytic fibrotic myocarditis suggestive of chronic canine CD (Fig. 4). All five dogs diagnosed in 2020 were PCR-positive for *T. cruzi* with DNA extracted from heart tissue. Three of these dogs were also positive with DNA extracted from spleen tissue. One pup, a four-month-old hound, was from a home located near the 18-month-old foxhound diagnosed with canine CD in 2019.

- A single canine Chagas diagnosis (case 11, Table) was made in 2021. The animal was a two-year-old mixed breed female that presented with paresis in her left forelimb for approximately two months preceding and progressing to quadriparesis. The animal was unable to walk and unresponsive to corticosteroid, NSAID and B vitamin therapy; the owner eventually decided to euthanize her. The dog resided in a household where over 80 triatomine bugs have been captured since 2016.

- The final four dogs were diagnosed with *T. cruzi* infection in 2023, and they were all from a single household. After viewing an educational flyer on CD distributed by our group to veterinary clinics throughout Trinidad, the dog owner discovered *T. cruzi* vector species *P. geniculatus* feeding on his sick 14-year-old German Shepard. The dog tested positive for *T. cruzi* infection and the owner subsequently requested that his four other dogs (all asymptomatic) be tested as well. Three of these four dogs were PCR-positive for *T. cruzi* infection, meaning that four of the five dogs from this household were infected with *T. cruzi*, although only one was symptomatic.

## DISCUSSION

*Trypanosoma cruzi* has been detected in Trinidad since 1958, but the parasite has not been studied in dogs on the island until now.[41] Here, we provide the first reports of locally transmitted canine *T. cruzi* infections. Severity between cases varied, but still conformed to the profile of canine CD. Three of the most severe cases occurred in pups under nine months old, which aligns with previous observations of younger dogs being more likely to show signs of acute disease.[22] Our findings suggest the existence of an active *T. cruzi* transmission cycle in Trinidad involving canines.

*Variation in clinical signs reflect challenges in canine Chagas surveillance and diagnosis* - Three of our youngest cases had non-specific symptoms that were initially mistaken for a different, more frequently observed condition, which is a common challenge in identifying canine CD.[48] Case 3, the five-week-old Cocker Spaniel pup, was initially thought to have tick fever and hookworm. Case 9, the three-month-old Husky pup that would not eat or drink, which was assumed to be a side effect of recent vaccinations. Unfortunately, both animals died within days of illness onset and necropsy examination revealed that both had had severe protozoal myocarditis. In addition, the symptoms of *T. cruzi* infection can be masked by a concomitant infection, which we observed in the four-month-old hound pup (case 8). The animal was severely infected with gastrointestinal helminths, which possibly

TABLE

Descriptions for the 15 canine *Trypanosoma cruzi* infection cases identified in Trinidad

| Case # | Year | Signalment | Origin | History | Gross | Histology | PCR + | Notes |
|---|---|---|---|---|---|---|---|---|
| 1* | 2008 | 4 year old, intact male, Dalmatian | Diego Martin | Sudden death | Moderate pulmonary oedema; pale heart | Severe necrotizing lymphoplasmacytic myocarditis with intracytoplasmic amastigotes | NA | |
| 2* | 2013 | 9 mo., intact female, Rottweiler | Maraval | Sudden death | No significant lesions | Severe necrotizing lymphoplasmacytic myocarditis with intracytoplasmic amastigotes | NA | |
| 3 | 2018 | 5 wk., intact male, Cocker Spaniel | Diego Martin | On medication for tick fever (doxycycline) enrocillina and Alban© for hookworm for two days before it died. | Moderate ascites; hydropericardium | Myocarditis severe, lymphoplasmocytic and histiocytic, necrotising with intracytoplasmic amastigotes | Blood | |
| 4+ | 2018 | Adult, intact female, Cocker Spaniel | Diego Martin | Dam to above | NA | NA | Blood | |
| 5 | 2019 | 1.5 yr old, intact male, foxhound | Cunaripo | Anorexia with marked weakness and hindlimb paresis. Euthanized. | *T. cruzi* forms visible in blood smear; pale heart muscle | Myocarditis with intracytoplasmic amastigotes. Moderate multifocal lymphoplasmocytic and monocytic lumbar myelitis | Spleen and heart | |
| 6 | 2020 | Adult, intact male, Hound | Not Stated | No Chagas symptoms; taken to clinic because it was hit by a car; luxation of T11 on spinal cord and commuted fracture of femur. Euthanized. | Fractures and splenic nodular hyperplasia | No cardiac lesions observed | Spleen and heart | |
| 7 | 2020 | Adult, intact female, Hound | Not Stated | Owner submitted carcass for necropsy. No history. | Multifocal moderate renal fibrosis and mild pulmonary oedema. | No cardiac lesions observed | Spleen and heart | |
| 8 | 2020 | 4 mo., intact female, Hound | Cunaripo | Submitted for necropsy. Pup was not eating and had pale mm. Swollen abdomen. | Haemorrhagic enteritis, hydrothorax and ascites, subcutaneous oedema. GIN eggs 3+ -severe endoparasitism | No cardiac lesions observed. | Spleen and heart | Lived near positive case from 2019. |
| 9 | 2020 | 3 mo., intact male, Husky | South Oropouche | Received 3rd round of routine vaccinations^ on 20/06/2020. Refused meal the next day. Started eating and drinking little. Was placed on oral rehydration, but continued to not eat or drink and died the same month (June). | Severe anaemia, mild hydrothorax and ascites | Myocarditis with myocardial degeneration and necrosis. Severe, acute, diffuse, with intra-myocytic protozoal amastigotes. | Heart and blood | |
| 10 | 2020 | Adult, intact male, Hound | Mamoral | Developed respiratory distress and died. | Pulmonary congestion and oedema, moderate and diffuse | Moderate multifocal lymphoplasmacytic and histiocytic fibrotic myocarditis. | Heart | From an armadillo hunting pack. |
| 11 | 2021 | Adult (2 yr), Spayed female, mixed breed | Coal Mine | Paresis in left forelimb for approximately 2 mo., progressing to quadriparesis. Unable to walk; unresponsive to corticosteroid, NSAID and B vitamin therapy. Euthanasized | NA | NA | Blood | From household where > 80 triatomine bugs captured since 2016 |
| 12+ | 2023 | Adult (14 yr), intact male, German shepherd | Manzanilla | Hindlimb paresis, urinary tract infection and prostatitis for several weeks. | NA | NA | Blood | Owner found triatomine bugs feeding on dog. |
| 13+ | 2023 | Adult (7 yr), Spayed female, mixed breed | Manzanilla | No clinical disease | | | Blood | From same yard as above |
| 14+ | 2023 | Adult (2 yr), Spayed female, Malinois | Manzanilla | No clinical disease | NA | NA | Blood | From same yard as above |
| 15+ | 2023 | Adult (5 yr), Spayed female, mixed breed | Manzanilla | No clinical disease | NA | NA | Blood | From same yard as above |

NA: indicates that procedure was not carried out; *Suspected cases (No polymerase chain reaction - PCR – confirmation); +Still alive at time of case record; ^Routine vaccinations in Trinidad consist of vaccines against the following: Canine distemper virus, canine adenovirus type 2, canine parainfluenza virus, canine parvovirus, and *Leptospira* spp.

masked her *T. cruzi* infection, which has been observed in other studies of *T. cruzi*-helminth coinfections.[49,50]

Another challenge with CD surveillance in both humans and dogs is that the chronic form of the disease often goes undiagnosed due to undetectable parasitemia or subclinical infection.[1,26,27] In our study, we observed several subclinical cases that would have gone undetected had it not been for other events prompting examination of the animals. For example, in the 2020 case of the hound from Mamoral (case 10), the animal had no observable *T. cruzi* forms even though the degree of fibrosis indicated

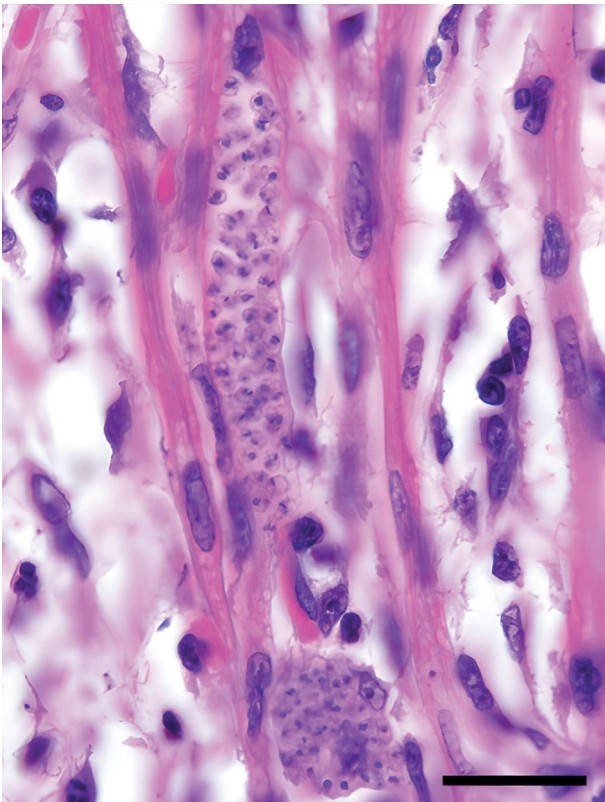

Fig. 2: myocarditis with intra-cardiomyocyte pseudocyst containing amastigotes from five-week-old Cocker Spaniel that died in 2018 (Case 4). Polymerase chain reaction (PCR) was *Trypanosoma cruzi* positive. Hematoxylin and Eosinstain (H&E) stain, 100x magnification. Bar = 20μm.

that the changes were chronic and PCR detected *T. cruzi* DNA. A second hound from 2020 (case 6) was taken to the clinic because it was hit by a car.

Finally, although CD presents with a wide range of symptoms, most studies of *T. cruzi* infection in dogs focus on cardiac manifestations, which may result in misdiagnoses in dogs displaying gastrointestinal (GI) or neurological alterations, which are reported in a small number of canine Chagas cases.[3,29,51] We found one dog with neurological signs (case 5, a foxhound with multifocal myelitis in 2019) although it was accompanied by classic signs of acute CD- circulating trypomastigotes, cardiac lesions, and positive PCR of the spleen and heart. No GI examinations were reported in the records we reviewed.

*Trypanosoma cruzi- infected mother and dam present possibility of transplacental T. cruzi transmission -* Our data included a *T. cruzi-* infected mother and pup (cases 4 and 3, respectively), presenting the possibility that the pup was infected with *T. cruzi* transplacentally. Vectorial transmission is also possible- the pup died at five weeks old, which is enough time for the pup to have acquired *T. cruzi* vectorially and developed peak parasitemia (~17 days post-infection).[22] As it stands, we have no direct evidence that clearly points to either transmission route and evidence of transplacental transmission is circumstantial. There are few studies of transplacental *T. cruzi* transmission in dogs, but one recent study of fetuses from naturally infected dams found a *T. cruzi* transmission frequency of 59%.[52] This transmission rate is much higher than that observed in humans, which has been estimated to be around 6%.[53,54] This is a notable difference given that the disease progression and pathology of CD in dogs and humans is otherwise considered to be quite similar.[1,3,23]

*Environmental factors -* Our findings suggest that in cases where nonspecific or unusual clinical pathology is observed, environmental factors of the dog's home and activities (*e.g.*, hunting, other outdoor activities, etc.) should be considered. The two hounds with histological lesions (cases 5 and 10, Table) were housed outdoors in kennels with multiple dogs kept close together, which are risk factors for canine CD.[19,20,55,56] The main prey of Case 10, a hunting hound from Mamoral, was armadillo, which is a *T. cruzi* reservoir that often cohabitates with triatomine bugs in its burrows.

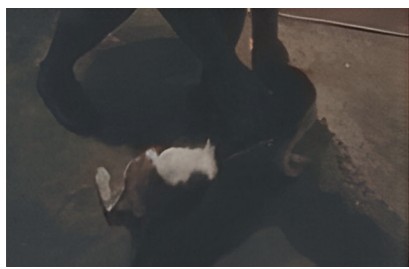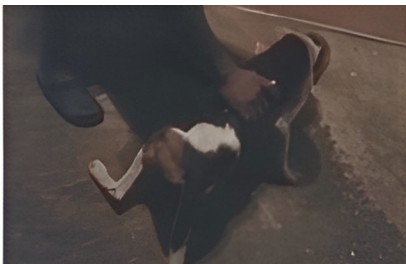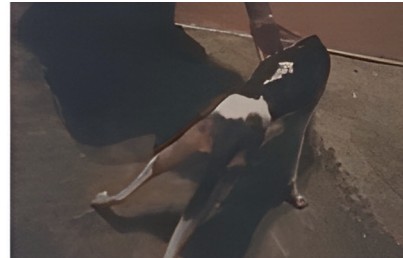

Fig. 3: stills from a video of the foxhound with limb paresis (Case 5). Given the presence of *Trypanosoma cruzi* amastigotes in the heart, parasite forms observed in the blood, and polymerase chain reaction (PCR)-positivity of the heart and spleen, the cause of the hindlimb paresis in the animal was likely Chagasic myelitis. However, direct evidence of *T. cruzi* in the spinal cord was not collected; PCR was conducted just with the heart and the spleen tissues, and immunohistochemistry was not carried out [Supplementary data (video)].

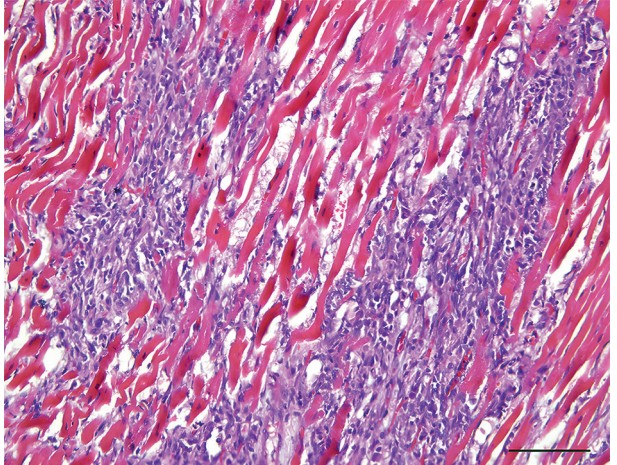

Fig. 4: chronic myocarditis with fibrosis from adult hound diagnosed in 2020 (Case 10). Heart tissue was polymerase chain reaction (PCR) positive for *Trypanosoma cruzi* infection. Hematoxylin and Eosin stain (H&E) stain, 20x magnification. Bar = 100μm.

The presence of triatomine bugs at a dog's home or residence are also important to consider, as highlighted by the mixed breed female from 2021 (case 11), which resided in a location where over 80 triatomine bugs have been collected since 2016. This animal presented with severe, nonspecific disease that did not conform to the classic Chagas pathology, yet whole blood from the dog tested positive for *T. cruzi*. In addition, triatomine bugs were found biting the dog from case 12 at its residence, which it shared with the dogs from cases 13-15. Taken together, these cases suggest active vector-borne *T. cruzi* transmission to canines occurring at their places of residence. Interestingly, the dog owner from cases 12-15 shared that he was bitten by a triatomine bug vector in the past and was unable to find any information about what had bit him. This anecdote highlights the need for publicly available information on both canine and human CD in Trinidad.[41]

*In conclusion* - Here, we provide evidence of locally acquired canine CD in Trinidad. As mentioned, dogs are considered a sentinel species for *T. cruzi* transmission to humans in some regions due to their close contact with people and propensity to consume insects.[10,17,57] Keeping in mind that canine *T. cruzi* infection does not extrapolate directly to human CD risk, our findings merit further research into this question as it plays out within the context of *T. cruzi* transmission in Trinidad. Moreover, our results highlight the need for increased CD surveillance and research in Trinidad and Tobago in order to estimate both human and veterinary risk and develop appropriate prevention and control strategies.

## ACKNOWLEDGEMENTS

To Gerald Chandoo for his help in this project. The authors dedicate this paper to the owners of the dogs reported in this manuscript.

## AUTHORS' CONTRIBUTION

RS, AM, AR and CO designed the study; RS, IP and LG executed the study; RS, IP, CO, VT and NLG analysed the data; RP and JKP wrote the manuscript drafts; RP, AM, AR, CO, NLG and JKP revised the manuscript. The authors declare that they have no competing interests.

## DATA AVAILABILITY

The contents underlying the research text are included in the manuscript.

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

# OPEN PEER REVIEW

Memórias do IOC thanks the anonymous reviewers for their contribution to the peer review of this work.

## FIRST REVIEW ROUND

REVIEWERS' COMMENTS

### REVIEWER #1

General Comments

This is an important contribution documenting the first evidence of locally acquired Trypanosoma cruzi infection in domestic dogs in Trinidad. The authors present data from nearly 4,000 diagnostic records spanning 15 years, identifying 13 confirmed and 2 suspected cases. The case descriptions are compelling, and the work has clear One Health implications for both veterinary and public health audiences.

The manuscript is generally well organized, and the writing is very clear, but several areas need clarification or refinement. The major issues largely concern presentation of data, interpretation of transmission routes, and a need for greater precision in some claims. In addition, the manuscript would benefit from editorial tightening to reduce redundancy and to improve clarity for an international readership.

Overall, I find this to be a strong manuscript that merits publication after revision.

Major Concerns

1. Line 251: The possibility of transplacental transmission is raised based on one dam–pup pair. Please note explicitly that congenital vs. vectorial infection cannot be distinguished without more direct evidence. I worry that the heading "Transplacental T. cruzi transmission" is going to be misleading to casual readers. If T. cruzi vectors were present and feeding on the dam, would it not be likely that they would also feed on her offspring? Would the prepatent period of T. cruzi make it possible that this puppy could have been infected by insect vectors? Please clarify in the text and add an element of significant caution to interpreting this is a possible transplacental infection. That said, the article reference does make a compelling argument for transplacental infection.

2. The conclusion (lines 280–285) calls for further research but could be more explicit about how canine cases may precede or signal underrecognized human cases. Consider emphasizing the sentinel role of dogs more strongly, while also clarifying limitations in extrapolating directly to human risk.

3. Figures and supplementary material - Some figure references (e.g., Fig. 1 line 99; Fig. 3 line 178) lack sufficient description in the text. Do we need to be able to recognize Panstrongylus geniculatus? If so, why? Fig. 3 is linked to a supplemental video, but the value added by this video is not clearly explained. How did the hindlimb paresis relate to the diagnosis of T. cruzi? Were CNS lesions attributed to T. cruzi found in this dog?

Line-specific edits

• Line 60: Typo: "such as such as raccoons…" Delete the duplicate text.

• Line 83: "Megaoesophagus and megacolon may also be present…"

• Consider citing veterinary-specific references in addition to human disease literature.

• Line 101: "upwards of 80%" "…. Up to 80%" is clearer and less colloquial.

• Line 127: "All cases were presumed to be locally acquired…" Suggest downgrading this statement to "All cases were likely locally acquired, as no history of travel outside the country was reported."

• Line 162: When numbers and units are used as a compound adjective they must be hyphenated (e.g. four-year-old, male Dalmatian)

• Line 214–217: This section notes "canines of multiple ages, sexes, and breeds." Consider removing or replacing with more specific epidemiologic observations. It seems to contradict the previous statement of younger dogs being more susceptibe.

• Line 222–224: Sentence beginning "Case 3…" is very long; consider creating two sentences for clarity.

• Line 273–277: The anecdote about the owner bitten by a triatomine bug is interesting but would be clearer if shortened. Currently it reads more like a case report within the Discussion.

• References: Ensure consistency of formatting (italics for species names, hyphenation in compound adjectives).

Strengths of the manuscript

• This is the first report of canine T. cruzi in Trinidad

• Large retrospective dataset across multiple institutions.

• Well-integrated clinical, pathologic, and molecular findings.

• Clear One Health relevance, raising awareness of neglected tropical disease in the Caribbean.

Recommendation

Minor Revision.

1. The manuscript is of significant value and should be published after the authors:

2. Moderate claims about transmission (especially congenital).

3. Improve clarity and conciseness in several sections.

With these revisions, the paper will provide a solid contribution to both veterinary parasitology and zoonotic disease literature.

## AUTHORS' RESPONSE TO THE REVIEWERS

REVIEWER COMMENTS:

Reviewer: 1

Reviewer comments:

General Comments

This is an important contribution documenting the first evidence of locally acquired Trypanosoma cruzi infection in domestic dogs in Trinidad. The authors present data from nearly 4,000 diagnostic records spanning 15 years, identifying 13 confirmed and 2 suspected cases. The case descriptions are compelling, and the work has clear One Health implications for both veterinary and public health audiences.

The manuscript is generally well organized, and the writing is very clear, but several areas need clarification or refinement. The major issues largely concern presentation of data, interpretation of transmission routes, and a need for greater precision in some claims. In addition, the manuscript would benefit from editorial tightening to reduce redundancy and to improve clarity for an international readership.

Overall, I find this to be a strong manuscript that merits publication after revision.

Major Concerns

1. Line 251: The possibility of transplacental transmission is raised based on one dam–pup pair. Please note explicitly that congenital vs. vectorial infection cannot be distinguished without more direct evidence. I worry that the heading "Transplacental T. cruzi transmission" is going to be misleading to casual readers. If T. cruzi vectors were present and feeding on the dam, would it not be likely that they would also feed on her offspring? Would the prepatent period of T. cruzi make it possible that this puppy could have been infected by insect vectors? Please clarify in the text and add an element of significant caution to interpreting this is a possible transplacental infection. That said, the article reference does make a compelling argument for transplacental infection. Thank you for this suggestion. We have added more cautious wording and changed the subtitle to reflect the lack of direct evidence for transplacental transmission (Lines 249-255).

2. The conclusion (lines 280–285) calls for further research but could be more explicit about how canine cases may precede or signal underrecognized human cases. Consider emphasizing the sentinel role of dogs more strongly, while also clarifying limitations in extrapolating directly to human risk. We have reworked the conclusion to emphasize more strongly the role of dogs as sentinels while also emphasizing that dogs don't guarantee human T. cruzi transmission and that this question must be investigated locally.

3. Figures and supplementary material - Some figure references (e.g., Fig. 1 line 99; Fig. 3 line 178) lack sufficient description in the text. Do we need to be able to recognize Panstrongylus geniculatus? If so, why? The reviewer raises a valid point- it is not necessary to recognize P. geniculatus to understand our study. We included the photo because the vector is under-studied in Trinidad, and there are not many locally available resources with pictures of P. geniculatus. Therefore, we felt it was important to include a photo of the vector for any veterinarians or dog owners in Trinidad or Tobago who might read this paper. However, if this is considered superfluous by the editorial staff and/or reviewers, we can remove it. Fig. 3 is linked to a supplemental video, but the value added by this video is not clearly explained. How did the hindlimb paresis relate to the diagnosis of T. cruzi? Were CNS lesions attributed to T. cruzi found in this dog? The hindlimb paresis was highly suggestive of chagasic myelitis given the findings of cardiac disease with T cruzi amastigotes and PCR-positive heart tissue. However, we do not have direct evidence of chagasic myelitis in the animal, which would need to be done with immunohistochemistry or PCR from the spinal cord. The video is included to add more detail, since the picture is not high resolution. We have adjusted the text to reflect that the hindlimb paresis was suspected of being due to chagasic myelitis given that the animal had T. cruzi in the heart, spleen, and blood, but that direct evidence was not collected to confirm the presence of T. cruzi in the spinal cord. Line-specific edits

• Line 60: Typo: "such as such as raccoons…" Delete the duplicate text. Done.

• Line 83: "Megaoesophagus and megacolon may also be present…" We are unsure of what is being requested from this sentence.

• Consider citing veterinary-specific references in addition to human disease literature. We are unsure which line is indicated for this suggestion; in general, the majority of our citations are canine-specific studies.

• Line 101: "upwards of 80%" "…. Up to 80%" is clearer and less colloquial. We have changed the wording to reflect the data, which is that over 80% of specimens were infected.

• Line 127: "All cases were presumed to be locally acquired…" Suggest downgrading this statement to "All cases were likely locally acquired, as no history of travel outside the country was reported." Done.

• Line 162: When numbers and units are used as a compound adjective they must be hyphenated (e.g. four-year-old, male Dalmatian) Done.

• Line 214–217: This section notes "canines of multiple ages, sexes, and breeds." Consider removing or replacing with more specific epidemiologic observations. It seems to contradict the previous statement of younger dogs being more susceptible. We have deleted this phrase as suggested.

• Line 222–224: Sentence beginning "Case 3…" is very long; consider creating two sentences for clarity. We have divided the phrase into two sentences.

• Line 273–277: The anecdote about the owner bitten by a triatomine bug is interesting but would be clearer if shortened. Currently it reads more like a case report within the Discussion. Thank you for the suggestion- we have made the anecdote more concise.

• References: Ensure consistency of formatting (italics for species names, hyphenation in compound adjectives). Done.

Strengths of the manuscript

• This is the first report of canine T. cruzi in Trinidad

• Large retrospective dataset across multiple institutions.

• Well-integrated clinical, pathologic, and molecular findings.

• Clear One Health relevance, raising awareness of neglected tropical disease in the Caribbean.

Recommendation Minor Revision.

1. The manuscript is of significant value and should be published after the authors: 2. Moderate claims about transmission (especially congenital). 3. Improve clarity and conciseness in several sections. With these revisions, the paper will provide a solid contribution to both veterinary parasitology and zoonotic disease literature.

## SECOND REVIEW ROUND

**REVIEWERS' COMMENTS**

### REVIEWER #1

No comments.

