## [Reviewer Report · FIRST REVIEW ROUND - REVIEWERS COMMENTS]

## REVIEWER #1

General Comments

This is an important contribution documenting the first evidence of locally acquired Trypanosoma cruzi infection in domestic dogs in Trinidad. The authors present data from nearly 4,000 diagnostic records spanning 15 years, identifying 13 confirmed and 2 suspected cases. The case descriptions are compelling, and the work has clear One Health implications for both veterinary and public health audiences.

The manuscript is generally well organized, and the writing is very clear, but several areas need clarification or refinement. The major issues largely concern presentation of data, interpretation of transmission routes, and a need for greater precision in some claims. In addition, the manuscript would benefit from editorial tightening to reduce redundancy and to improve clarity for an international readership.

Overall, I find this to be a strong manuscript that merits publication after revision.

Major Concerns

1. Line 251: The possibility of transplacental transmission is raised based on one dam–pup pair. Please note explicitly that congenital vs. vectorial infection cannot be distinguished without more direct evidence. I worry that the heading “Transplacental T. cruzi transmission” is going to be misleading to casual readers. If T. cruzi vectors were present and feeding on the dam, would it not be likely that they would also feed on her offspring? Would the prepatent period of T. cruzi make it possible that this puppy could have been infected by insect vectors? Please clarify in the text and add an element of significant caution to interpreting this is a possible transplacental infection. That said, the article reference does make a compelling argument for transplacental infection.

2. The conclusion (lines 280–285) calls for further research but could be more explicit about how canine cases may precede or signal underrecognized human cases. Consider emphasizing the sentinel role of dogs more strongly, while also clarifying limitations in extrapolating directly to human risk.

3. Figures and supplementary material - Some figure references (e.g., Fig. 1 line 99; Fig. 3 line 178) lack sufficient description in the text. Do we need to be able to recognize Panstrongylus geniculatus? If so, why? Fig. 3 is linked to a supplemental video, but the value added by this video is not clearly explained. How did the hindlimb paresis relate to the diagnosis of T. cruzi? Were CNS lesions attributed to T. cruzi found in this dog?

Line-specific edits

• Line 60: Typo: “such as such as raccoons…” Delete the duplicate text.

• Line 83: “Megaoesophagus and megacolon may also be present…”

• Consider citing veterinary-specific references in addition to human disease literature.

• Line 101: “upwards of 80%” “…. Up to 80%” is clearer and less colloquial.

• Line 127: “All cases were presumed to be locally acquired…” Suggest downgrading this statement to “All cases were likely locally acquired, as no history of travel outside the country was reported.”

• Line 162: When numbers and units are used as a compound adjective they must be hyphenated (e.g. four-year-old, male Dalmatian)

• Line 214–217: This section notes “canines of multiple ages, sexes, and breeds.” Consider removing or replacing with more specific epidemiologic observations. It seems to contradict the previous statement of younger dogs being more susceptibe.

• Line 222–224: Sentence beginning “Case 3…” is very long; consider creating two sentences for clarity.

• Line 273–277: The anecdote about the owner bitten by a triatomine bug is interesting but would be clearer if shortened. Currently it reads more like a case report within the Discussion.

• References: Ensure consistency of formatting (italics for species names, hyphenation in compound adjectives).

Strengths of the manuscript

• This is the first report of canine T. cruzi in Trinidad

• Large retrospective dataset across multiple institutions.

• Well-integrated clinical, pathologic, and molecular findings.

• Clear One Health relevance, raising awareness of neglected tropical disease in the Caribbean.

Recommendation

Minor Revision.

1. The manuscript is of significant value and should be published after the authors:

2. Moderate claims about transmission (especially congenital).

3. Improve clarity and conciseness in several sections.

With these revisions, the paper will provide a solid contribution to both veterinary parasitology and zoonotic disease literature.

---

## [Author Response · AUTHORS RESPONSE TO REVIEWERS]

## REVIEWER COMMENTS:

Reviewer: 1

Reviewer comments:

General Comments

This is an important contribution documenting the first evidence of locally acquired Trypanosoma cruzi infection in domestic dogs in Trinidad. The authors present data from nearly 4,000 diagnostic records spanning 15 years, identifying 13 confirmed and 2 suspected cases. The case descriptions are compelling, and the work has clear One Health implications for both veterinary and public health audiences.

The manuscript is generally well organized, and the writing is very clear, but several areas need clarification or refinement. The major issues largely concern presentation of data, interpretation of transmission routes, and a need for greater precision in some claims. In addition, the manuscript would benefit from editorial tightening to reduce redundancy and to improve clarity for an international readership.

Overall, I find this to be a strong manuscript that merits publication after revision.

Major Concerns

1. Line 251: The possibility of transplacental transmission is raised based on one dam–pup pair. Please note explicitly that congenital vs. vectorial infection cannot be distinguished without more direct evidence. I worry that the heading “Transplacental T. cruzi transmission” is going to be misleading to casual readers. If T. cruzi vectors were present and feeding on the dam, would it not be likely that they would also feed on her offspring? Would the prepatent period of T. cruzi make it possible that this puppy could have been infected by insect vectors? Please clarify in the text and add an element of significant caution to interpreting this is a possible transplacental infection. That said, the article reference does make a compelling argument for transplacental infection.

Thank you for this suggestion. We have added more cautious wording and changed the subtitle to reflect the lack of direct evidence for transplacental transmission (Lines 249-255).

2. The conclusion (lines 280–285) calls for further research but could be more explicit about how canine cases may precede or signal underrecognized human cases. Consider emphasizing the sentinel role of dogs more strongly, while also clarifying limitations in extrapolating directly to human risk.

We have reworked the conclusion to emphasize more strongly the role of dogs as sentinels while also emphasizing that dogs don’t guarantee human T. cruzi transmission and that this question must be investigated locally.

3. Figures and supplementary material - Some figure references (e.g., Fig. 1 line 99; Fig. 3 line 178) lack sufficient description in the text. Do we need to be able to recognize Panstrongylus geniculatus? If so, why?

The reviewer raises a valid point- it is not necessary to recognize P. geniculatus to understand our study. We included the photo because the vector is under-studied in Trinidad, and there are not many locally available resources with pictures of P. geniculatus. Therefore, we felt it was important to include a photo of the vector for any veterinarians or dog owners in Trinidad or Tobago who might read this paper. However, if this is considered superfluous by the editorial staff and/or reviewers, we can remove it.

Fig. 3 is linked to a supplemental video, but the value added by this video is not clearly explained. How did the hindlimb paresis relate to the diagnosis of T. cruzi? Were CNS lesions attributed to T. cruzi found in this dog?

The hindlimb paresis was highly suggestive of chagasic myelitis given the findings of cardiac disease with T cruzi amastigotes and PCR-positive heart tissue. However, we do not have direct evidence of chagasic myelitis in the animal, which would need to be done with immunohistochemistry or PCR from the spinal cord. The video is included to add more detail, since the picture is not high resolution. We have adjusted the text to reflect that the hindlimb paresis was suspected of being due to chagasic myelitis given that the animal had T. cruzi in the heart, spleen, and blood, but that direct evidence was not collected to confirm the presence of T. cruzi in the spinal cord.

Line-specific edits

• Line 60: Typo: “such as such as raccoons…” Delete the duplicate text.

Done.

• Line 83: “Megaoesophagus and megacolon may also be present…”

We are unsure of what is being requested from this sentence.

• Consider citing veterinary-specific references in addition to human disease literature.

We are unsure which line is indicated for this suggestion; in general, the majority of our citations are canine-specific studies.

• Line 101: “upwards of 80%” “…. Up to 80%” is clearer and less colloquial.

We have changed the wording to reflect the data, which is that over 80% of specimens were infected.

• Line 127: “All cases were presumed to be locally acquired…” Suggest downgrading this statement to “All cases were likely locally acquired, as no history of travel outside the country was reported.”

Done.

• Line 162: When numbers and units are used as a compound adjective they must be hyphenated (e.g. four-year-old, male Dalmatian)

Done.

• Line 214–217: This section notes “canines of multiple ages, sexes, and breeds.” Consider removing or replacing with more specific epidemiologic observations. It seems to contradict the previous statement of younger dogs being more susceptible.

We have deleted this phrase as suggested.

• Line 222–224: Sentence beginning “Case 3…” is very long; consider creating two sentences for clarity.

We have divided the phrase into two sentences.

• Line 273–277: The anecdote about the owner bitten by a triatomine bug is interesting but would be clearer if shortened. Currently it reads more like a case report within the Discussion.

Thank you for the suggestion- we have made the anecdote more concise.

• References: Ensure consistency of formatting (italics for species names, hyphenation in compound adjectives).

Done.

Strengths of the manuscript

• This is the first report of canine T. cruzi in Trinidad

• Large retrospective dataset across multiple institutions.

• Well-integrated clinical, pathologic, and molecular findings.

• Clear One Health relevance, raising awareness of neglected tropical disease in the Caribbean.

Recommendation Minor Revision.

1. The manuscript is of significant value and should be published after the authors:

2. Moderate claims about transmission (especially congenital).

3. Improve clarity and conciseness in several sections.

With these revisions, the paper will provide a solid contribution to both veterinary parasitology and zoonotic disease literature.